# The Architecture Design of Electrical Vehicle Infrastructure Using Viable System Model Approach

**Mahdi Boucetta** [1]**, Niamat Ullah Ibne Hossain** [1]**, Raed Jaradat** [1,*]**, Charles Keating** [2]**, Siham Tazzit** [1]
**and Morteza Nagahi** [1]

1   Department of Industrial and System Engineering, Mississippi State University,
    Mississippi State, MS 39762, USA; mb3736@msstate.edu (M.B.); ni78@msstate.edu (N.U.I.H.);
    st1628@msstate.edu (S.T.); mn852@msstate.edu (M.N.)
2   Department of Engineering Management & Systems Engineering, Old Dominion University,
    Norfolk, VA 23508, USA; ckeating@odu.edu
*   Correspondence: jaradat@ise.msstate.edu

**Abstract:** Exponential technological-based growth in industrialization and urbanization, and the ease of mobility that modern motorization offers have significantly transformed social structures and living standards. As a result, electric vehicles (EVs) have gained widespread popularity as a mode of sustainable transport. The increasing demand for of electric vehicles (EVs) has reduced the some of the environmental issues and urban space requirements for parking and road usage. The current body of EV literature is replete with different optimization and empirical approaches pertaining to the design and analysis of the EV ecosystem; however, probing the EV ecosystem from a management perspective has not been analyzed. To address this gap, this paper develops a systems-based framework to offer rigorous design and analysis of the EV ecosystem, with a focus on charging station location problems. The study framework includes: (1) examination of the EV charging station location problem through the lens of a systems perspective; (2) a systems view of EV ecosystem structure; and (3) development of a reference model for EV charging stations by adopting the viable system model. The paper concludes with the methodological implications and utility of the reference model to offer managerial insights for practitioners and stakeholders.

**Keywords:** EV ecosystem; EV charging station; viable system model; cybernetics; management





## 1. Introduction

In the last decade, the passenger and freight transport sector has become a focus of attention as statistics show this sector has the highest emissions of greenhouse gases. In particular, road transport is the main emitter of $CO_2$, with 28.9% of total U.S emissions in 2017 [1]. Beyond the reduction of greenhouse gas emissions, policies and regulations are also aimed at improving air quality, which is considered a major public health concern. The management of greenhouse gases and air pollution must be controlled simultaneity since identical fuels are responsible for these types of emissions. To reduce greenhouse gas emissions and consumption of fossil fuels, and to improve air quality and engage strategies for a post-oil economy, many global industries have shifted focus on the development of clean transport and low-carbon mobility [2,3]. With this shift, the movement towards electrification of mobility is gaining strength as part of greening transportation systems.

As an alternative to fossil-fuel vehicles, electric vehicles (EVs) have become of interest to mitigate resource scarcity and reduce environmental deterioration [2–4]. EVs have the potential to change the structure of the vehicle-manufacturing sector as well as the current fuel-vehicle ecosystem. EVs include an array of alternatives, including plug-in electric cars, hybrid electric cars, hydrogen vehicles, electric trains, electric trucks, and electric motorcycles/scooters. Although several regions across the globe incorporate different strategies to adopt EV technology, the current support infrastructure is not adequate.

For example, some regions struggle to understand how EVs will complement/replace their existing transportation architecture. The transportation ecosystem provides the technologies, services, and processes necessary to facilitate market penetration and includes both public and private actors, with private actors being more dominant. One of the main challenges currently facing EVs is the optimization of suitable charging locations. Between the immature development of charging point networks and the low capacity of current batteries, EVs still encounter the chicken and egg dilemma. The management and development of EVs and recharging infrastructures is a necessity to address this dilemma. This paper addresses the current challenge by presenting a more systemic approach for the EV charging station locations through the establishment of an EV conceptual model, the examination of the related stakeholders and actors, and the identification of key challenges pertaining to charging stations. The viable system model (VSM) is used to suggest a more collaborative solution that provides synergy between EV sub-systems and the roles of stakeholders. The ability of the VSM to identify the sources of dysfunction in the design, operation, and development of a system can provide EVs ecosystem with the ability to achieve sustainable delivery of products and services. Since the VSM is grounded on systems theory and management cybernetics, it can provide fruitful insights to understand the EVs systemic issues through the functions of the five systems (S1–S5).

## 2. Related Research

To minimize environmental impact and gas emissions, the electric vehicle (EV) concept has gain attention in the last decade. To adopt the concept and to examine the operational feasibility of EVs, different researchers have utilized a wide range of techniques. A large number of studies have demonstrated the efficacy of operations research-based techniques such as stochastic modeling, integer programming, multilayer programming, linear/nonlinear programming, and multi-criteria decision-making (MCDM) methods in the context of different EV-related issues. For instance, [5] applied an MCDM technique to select the optimal sites for EV charging stations in China. They developed a set of evaluation metrics based on three criteria, including environmental, economic, and societal, to determine the optimal sites for electric vehicle charging stations (ECVSs). Similarly, [6] and [7] leveraged the fuzzy analytical hierarchy process (AHP) to pursue the same objective. In another study, [8] presented a linear integer model for the optimal location of EV charging stations by considering the minimization of the total travel cost from the demand point to the charging station location. Sadehi-Barzani et al. [9] conducted a study for optimal placing and sizing of the fast-charging EV stations by adopting a mixed-integer non-linear (MINLP) optimization approach. A hierarchical optimization model for a network of EV charging stations was proposed by [10]. They validated their three-layered systems model of fast charging stations (FCSs) using the state of Arizona and North Dakota highway network with a gravity data model. The output of the model indicated that their proposed model enhanced the system performance and quality of service of these states. In a different study, [11] developed a mathematical model to optimize the location-scheduling problem by proposing a private network for EV charging infrastructure in an urban area. A mixed-integer nonlinear programming-based approach was applied by [12] to optimize the EV charging station location problem based on stochastic travel distance. They proposed a Benders-and-Price algorithm to develop the model, which was validated through real-life experiments on the Texas highway network. Moving away from operation research techniques, simulation-based approaches have also been adopted in different domains of EVs. Interested readers can refer to the works of [13–19] for other notable efforts to investigate EVs. It is apparent from the thread of discussion that none of the studies have attempted to apply a more systemic (holistic) approach to address the EV charging station problem. Given the current state and challenges faced by current EV stakeholders, this is an appropriate time for such an examination.

Several other studies have focused on EV battery management, especially the optimization of routing and decisions regarding battery-swapping stations where an EV can

quickly interchange its depleted battery with a fully charged battery. The authors of [20] proposed a dynamic programming model to determine the optimal number of batteries a swap station should keep in stock to avoid excessive electricity cost. They selected the San Francisco Bay area as a testing ground to validate their model. In [21], the authors present a pair of optimization models that aided in strategic financial planning for establishing battery swapping infrastructure under demand uncertainty. The general insight drawn from the research reveals that the optimal control of charging station and battery-swapping demands is critical to managing and promoting agile planning. This planning, in conjunction with advanced technology, is necessary to establish optimal infrastructure systems for EVs. In another study, [22] modeled and analyzed the EV battery-swap-stations problem by adopting a dynamic programming technique. The results indicated that transshipment of batteries between stations and incentivizing the customer to shift their demand could substantially increase the stability of the entire EV battery swap station network.

Another stream of research focuses on the interoperability between EVs and the electric grid. For instance, [23] described different control methods (e.g., grid-to-vehicle and V2G) that balance the bidirectional communications between the EV and power grid. Two dynamic programming algorithms were proposed by [24] that aid in avoiding electric grid overload. The first algorithm optimizes charging time to reduce electricity costs, while the second algorithm incorporates vehicle-to-grid (V2G) energy transfer to support the grid. A somewhat different approach was followed by [25]. Their study demonstrated how parked EVs could be used as supplementary energy storage devices when plugged into the grid to meet the additional demands for electricity. Table 1 provides a summary of the current themes related to the different aspects of EVs. These themes serve as a baseline in the development of the proposed VSM model.

**Table 1.** Current themes of the electric vehicle (EV) literature.

| Authors | Area of Research | Approach |
|---|---|---|
| Guo and Zhao (2015) [5] | Optimal location for EV charging stations | Fuzzy TOPSIS |
| Ju et al. (2018) [6] | Optimal location for EV charging station | Analytical hierarchy process (AHP) |
| Erbas et al. (2018) [7] | Optimal location for EV charging station | Fuzzy TOPSIS and AHP |
| Frade et al. (2011) [26] | EV charging station | Optimization approach |
| Chen et al. (2013) [27] | EV charging station | Mixed-integer programming problem |
| Sadeghi-Barzani (2014) [9] | Optimal location for EV charging station | Mixed-integer non-linear (MINLP) optimization |
| Kong et al. (2017) [10] | Optimal location for EV charging stations | Hierarchical optimization model |
| Baouche et al. (2014) [8] | EV charging station | Optimization approach |
| Xu et al. (2013) [28] | Optimal layout for EV charging stations | Optimization approach |
| Dashora et al. (2010) [29] | Optimal planning for EV charging Station | Mixed-integer mathematical programming |
| Rezai et al. (2015) [25] | Demand response using aggregated PEVs in parking lots (Interoperability) | Multistage optimization approach |
| Pashajavid and Golkar (2013) [30] | Placement and sizing EV of charging station | Particle swarm optimization (PSO) approach |
| Metz and Doetsch (2012) [15] | Relationship between EV mobility and grid support | Simulation approach |
| Coninx and Holvoet (2014) [16] | EV Online charging | Simulation approach |
| Liu et al. (2015) [23] | Interoperability between V2G | Simulation approach |
| Dong et al. (2014) [31] | Optimal planning for EV charging Station | Genetic algorithm approach |
| Lee and Park (2015) [32] | Dual battery management for EV | A genetic algorithm approach |
| Cai et al. (2014) [33] | Optimal layout of EV charging stations | Big-data analytics |

Based on the existing literature, the general thread pertaining to the electric vehicle charging station (EVCS) can be clustered into four main categories. These categories include, the selection of the optimal location and size of ECVS, the determination of an optimal allocation of ECVSs, the selection of the optimal layout for the EVCS, and the development of optimal planning. It is also apparent from the literature that the selection of optimal location and size of ECVSs is the most frequent category.

Although there are some theoretical and analytical studies focused on solving different issues pertaining to EVs, there is scant research that has attempted to address the managerial aspects related to the EV charging station location problem. To address this gap, this paper develops a systems-based framework to more rigorously analyze the electric vehicle charging station problem from managerial and methodological perspectives. The proposed framework is based on systems theory laws and principles and management cybernetics, and it is illustrated in terms of the viable system model (VSM) pioneered by [34]. The development of the framework encompasses the following aspects:

- Framing the EV charging station location problem from a systems perspective;
- Articulation of a systems-based framework for the charging station location problem based on management cybernetics and systems theory;
- Supporting a more robust exploration of the EV charging station location problem while serving as a 'point of reflection' to provide recommendations that might be fruitful to the future development of EVs.

To the best of our knowledge, this is the first attempt to address the EV charging station problem from a systemic (holistic) perspective using the VSM. This study also presents the efficacy and extensibility of the VSM approach in the context of transportation and logistics management problems. The VSM has been used in different domains and applications such as project management [35–37], information systems [38–40], sustainability management [41–44], production and manufacturing systems [45–47], energy management [48], and education systems [49].

The remaining organization of this paper is as follows: Section 3 provides a comprehensive description of the viable system model (VSM), serving as the basis for systemic framing of the EVs ecosystem. Section 4 explores the optimal locations for EV charging stations by performing a stakeholder analysis that comprises the primary contributors of the EV ecosystem. We used the California state EVCS as a demonstrative case study to show the applicability of the VSM model.

## 3. Viable System Model

In the 1950s, Stafford Beer developed the viable system model (VSM) as an approach to solving the organizational problems that traditional methods were not capable of solving [50]. Beer defined the VSM to be a "holistic model involving the intricate interactions of five identifiable but not separate sub-systems" [34]. The VSM offers the possibility of designing an organization as a system. This includes regulatory, adaptive, and learning capabilities to ensure the system will be robust in response to changes occurring in the system or environment, even though these changes may not have been detectable in the original system design.

The essence of the VSM is captured in system viability, which is the ability to maintain a separate existence. However, it is noteworthy that this viability does not assure effectiveness, only continued existence [51] The VSM permits the system structure to be decomposed into relationships between system entities that communicate by providing information flow within the system. The underlying structure of the VSM model is illustrated in Figure 1.

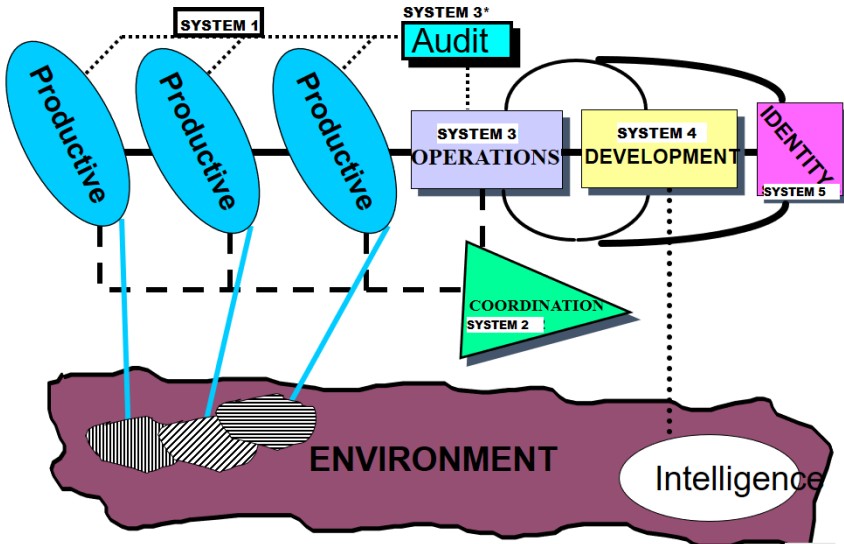

**Figure 1.** The underlying structure of the viable system model (VSM) model (Adapted from [34]).

The following development of the VSM is based on the works of [34]. Figure 1 represents the main existing systems (functions) that must be performed by every viable system. To maintain viability, the VSM suggests that the structure must be capable of performing the functions designated as systems one through five. The VSM seeks a design that maximizes autonomy (freedom and independence of decision, action, and interpretation) by the productive functions (System 1s) that produce the value of the system. The five VSM functions provide the necessary and sufficient conditions to deal with the turbulence in the system environment. System 1 facilitates the production of value by the system (e.g., products, services, and information). In other words, System 1 delivers system value which is provided as outputs that are consumed external to the system. The management (i.e., metasystem) consists of Systems 2, 3, 4, and 5; the interactions among these systems are achieved through mechanisms (i.e., vehicles providing interaction). The role of the metasystem is to provide cohesion within the whole system by ensuring that the operational units work together in a harmonious mode. The metasystem can be further divided into three main functions:

- The internal eye: Systems 2, 3, and 3 * are focused on the internal system. This ensures that the 'here and now' focus and responsibilities of the systems are executed effectively.
- The external eye: System 4 is focused externally and to the future on the "outside and then". It scans the environment and develops plans in the context of the outside world, trends, patterns, and emergent conditions and their implications for the future of the system.
- Policy system: System 5 develops, propagates, and maintains the identity of the system. It balances the emphasis between the System 3's present focus and the System 4's future focus to ensure that the system not only fulfills present requirements but also considers development that must be engaged to foster future viability based on changing environmental conditions.

From a holistic view, all functions are required to be performed to maintain the viability (both present and future existence) of the system. Communications is a central element of the VSM and involves the flow and interpretation of information within and external to the system. This communication is achieved through channels that guide structural interrelationships as the different functions are performed by a viable system. The fundamentals of the VSM model are summarized below, and a concise description of the role of each system (function) is discussed in Table 2. In sum, the central tenets of the VSM include:

- System viability is the ability to maintain a separate existence, and this existence does not assure effectiveness.
- All systems, natural or manmade, perform basic system functions.
- System structure is a function of relationships between system entities.
- Communications provide information flow and interpretation within the system.

**Table 2.** A concise description of each system (function) within the VSM.

| System Type | Primary Functions |
|---|---|
| System 1: Operations | Characterizes the operational units and manages the various production elements such as products, services, or information. This system also incorporates settings (i.e., requirements) to maintain the system's purpose and existence. |
| System 2: Coordination | Focuses on the role of coordination to ensure viability by solving the conflict between operational units and preventing unnecessary oscillations. |
| System 3: Control | Manages the performance level of the operational units. System 3 is responsible for defining directives, allocating resources, and establishing accountability for each operational unit. |
| System 3 *: Audit (monitoring) | Allows managers to audit performance without relying on the information they sent through System 2 and central channels connecting with System 3. These monitoring activities make the overall shared information more reliable. |
| System 4: Development | Predicts the future and diagnoses potential risks. Changes in the environment are detected and analyzed according to the system's main objectives. System 4 provides a set of action recommendations to ensure continued system viability in the face of environmental shifts. |
| System 5: Identity | Formulates the principles and goals of the system to provide consistency in the vision, mission, and purpose of the overarching system. This function ensures the preservation of the system's identity as it adapts to the changes that have occurred. |

## 4. Optimal Location Problem of EV Charging Station

The availability of the charging infrastructure for EVs can be associated with autonomous performance. If there were more charging points available, it would increase the autonomy of EVs to move in a less restrictive fashion. However, the challenges are not limited to this charging point availability. Consumers always consider extended charging time. With regard to the technical aspects of charging, some problems arise because of the current charging infrastructure, the battery technology, or the EV itself. In their research, [52] summarized the important determinants necessary to achieve customer acceptance. In the process of resolving the chicken and egg problem for EVs, it is necessary to build a prioritization hierarchy for EV challenges. Ultimately, the confluence of these priorities will be influential in securing consumer acceptance.

The energy distribution management within charging stations is considered another major concern for stakeholders, especially during peak hours. The smart grid could be a possible solution to this concern [53]. Vehicle-to-grid (V2G) technologies have also been developed to alleviate excessive loads on the power grid by planning optimal charging schedules for EVs [54]. Nonetheless, another serious problem or bottleneck for EVs is found in their batteries. Several issues related to battery technology need to be addressed, such as reducing weight, volume, charging times, dependence on operating temperature, and the use and treatment of toxic components. The customer acceptance variables of EVs are depicted in Figure 2, and technical aspects associated with EV infrastructure are illustrated in Figure 3.

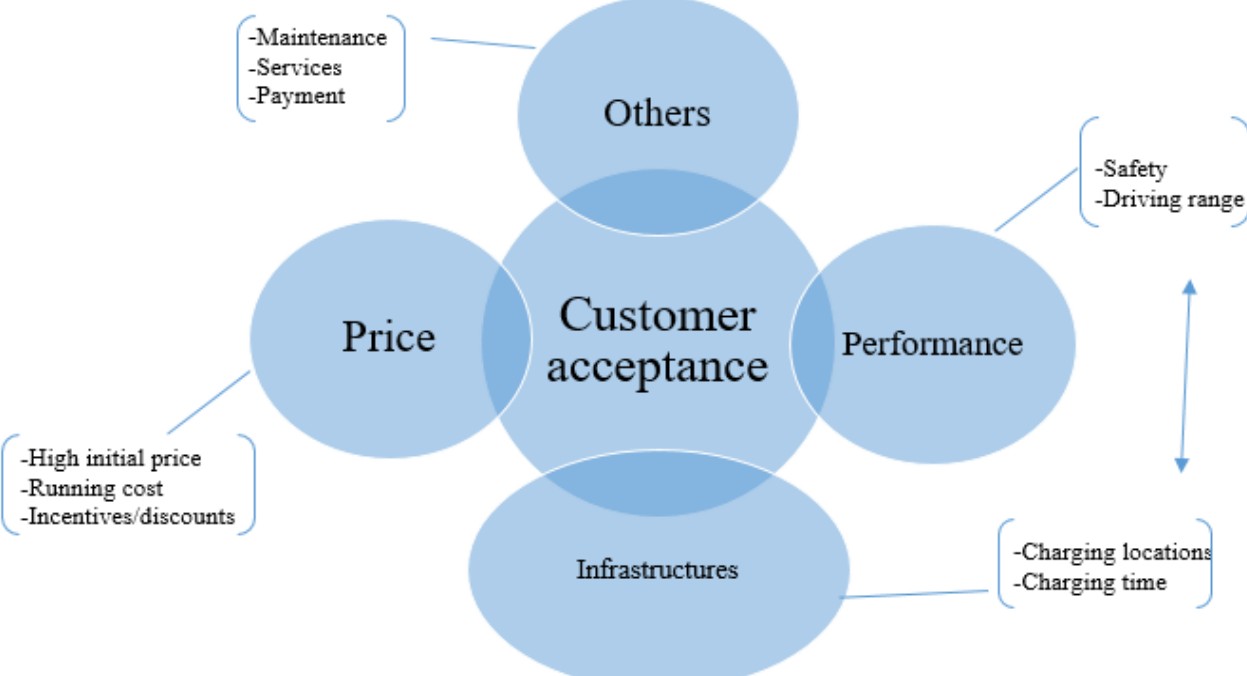

**Figure 2.** Customer acceptance variables.

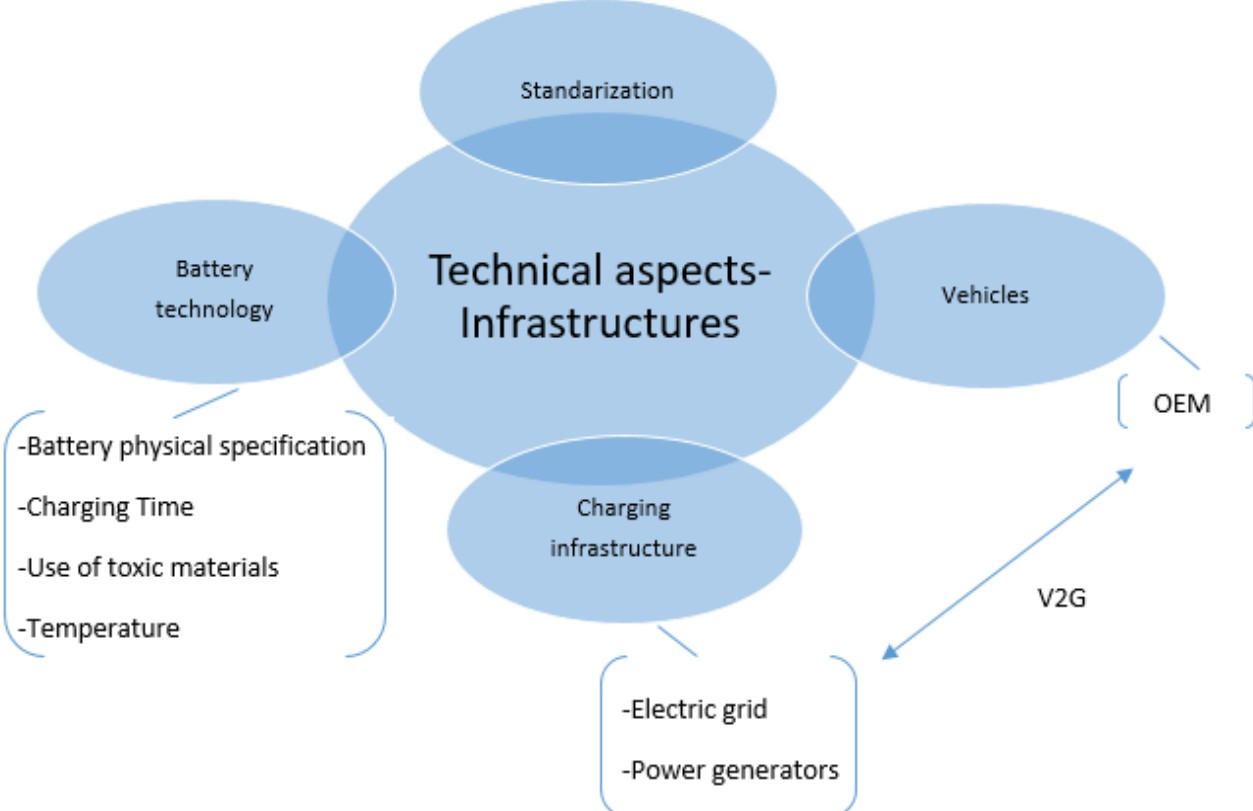

**Figure 3.** Technical aspects associated with EV infrastructure.

In the EV environment, the main stakeholders include government and policymakers, EV designers and manufacturers, suppliers, energy providers, users, and funding shareholders [55]. Figure 4 defines the relationship between the EVs stakeholders and actors within the transportation ecosystem, and Figure 5 illustrates electromobility main actors.

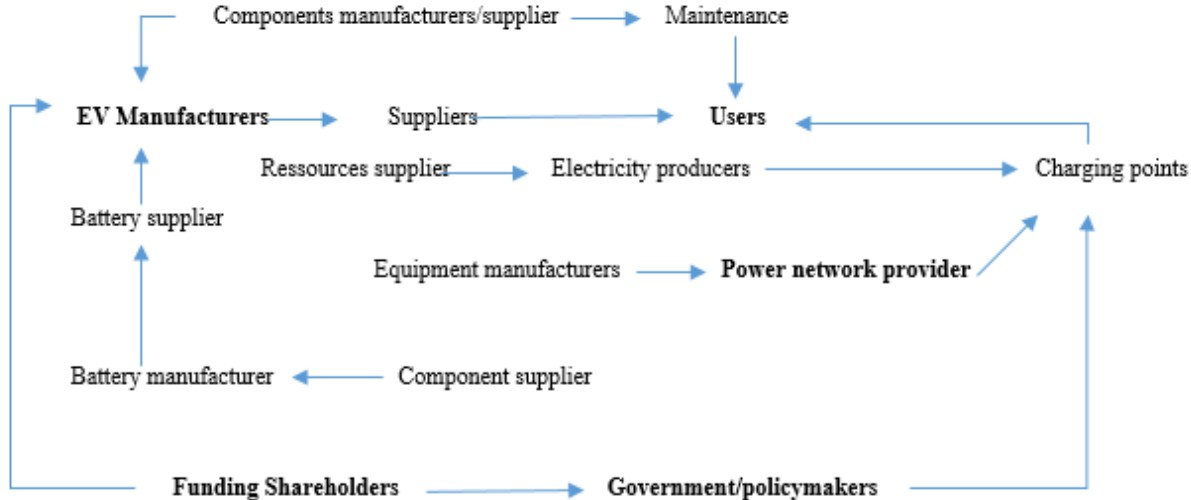

**Figure 4.** Electric vehicle ecosystem.

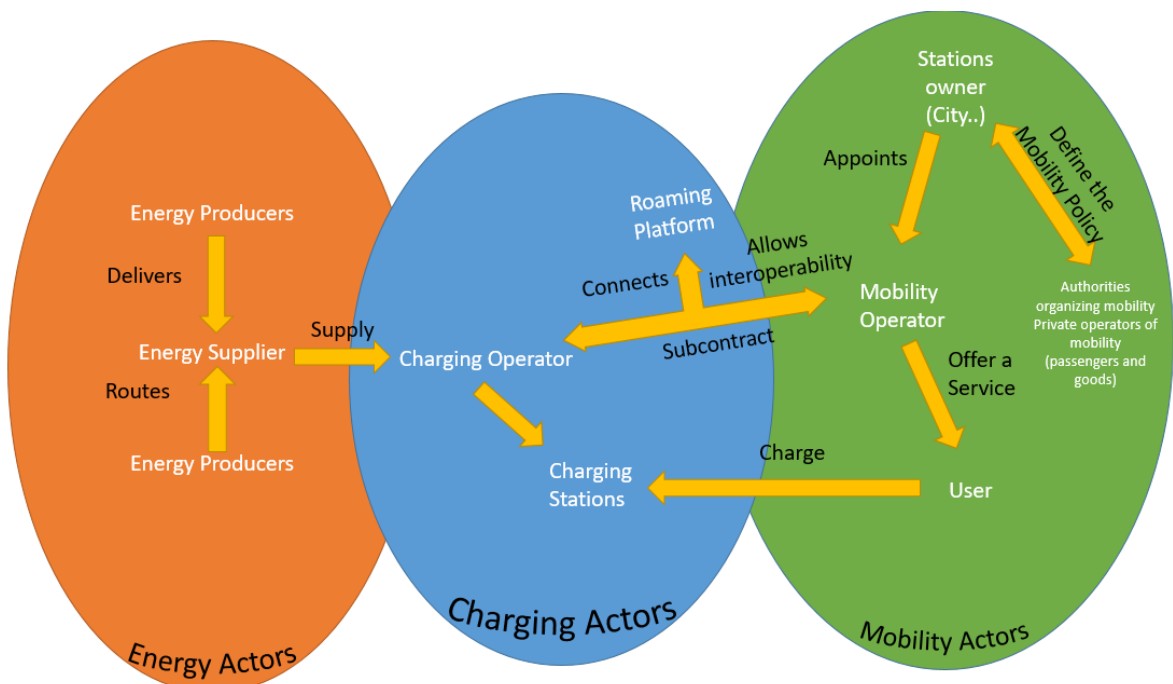

**Figure 5.** Electromobility main actors.

- Users are considered the largest stakeholder segment. They consist of consumers; therefore, consumer acceptance is critical to EVs.
- The energy sector consists of power providers and infrastructures (i.e., charging points).
- EV manufacturers play a major role in addressing the segmentation of EVs future, and comprise original equipment manufacturers (OEM), battery suppliers, and component suppliers.
- Battery manufacturers or battery suppliers play a major role together with the energy sector and infrastructure and EV manufacturers to address technical challenges.
- Government and policymakers include policymakers and regulators from any level of government, e.g., federal/state government, county, city, lobbyists, and interest groups with perceived EV interests.

- Funding shareholders, described as a small group with an influential investment stake, have overlapping areas of concern and collaboration with the government and policymakers.
- Energy network providers are in charge of conveying electricity from the production facilities to the consumer installations. There are two kinds of providers: (i) those that operate, maintain, and develop high voltage (HT) and very high voltage (HST) power lines that carry electricity from production units to the electricity distribution network and industrial customers; and (ii) those in charge of conveying energy from the transformer stations to the final consumer.
- The energy supplier supplies the site on which the charging stations are installed.
- The charging operator takes care of the technical operations of charging stations (maintenance and technical assistance).
- The mobility operator offers a charging service to its customers, which can regroup the networks of several charging operators. They are in contract with the electric vehicle user and have agreements with the roaming platform.
- The roaming platform is a platform for data exchange between mobile operators and charging operators. The roaming platform allows customers of a mobility operator to have access to all charging networks and thus enables the existence of interoperable networks. For example, GIREVE (group for the roaming of electric vehicle refills) is one of the roaming platforms on the French market. The roaming platform also identifies existing charging infrastructures and provides a terminal location and information service in real-time.

## 5. Integrative Model for EVCS Problem: A Case Study

Based on a recent study published by [56], the EV market in the US represents 22% of the global fleet, which translates to 360,000 units due to the recent efforts of the major EV manufacturers (mainly Tesla). According to the International Council on Clean Transportation (ICCT), the U.S is considered as the third largest EV market with more than 2 million EVs sold in 2018, whereas California represents half of the U.S electric vehicles market in 2017 [57].

California, along with other states, is continuing in the process of electrification. The authors of [58] quantified the gap in charging infrastructures from 2017 to what is needed in 2025, concluding that only 25% of the workplace and public chargers needed by 2025 are in place [59]. This gap is explained by the percentage increase for EV car sales compared to the percentage increase for implemented infrastructure. We chose the state of California as a case study to validate our model due to the EV emphasis and involvement of the state and local governments, as well as the size of the Californian EV market compared to the overall U.S market. California is among the few states that offer a model in the electrification process in the U.S due to the numerous policies and consumer incentives. Charging infrastructure is key to supporting EV market growth in the Californian example. Infrastructures represent 31% of all the U.S charging locations, as shown in Table 3 [59].

**Table 3.** Comparison of electric vehicles in California and the United States.

|  | US | California | California as Percent of U.S. |
|---|---|---|---|
| New 2017 electric vehicles | 193,000 | 96,000 | 50% |
| Cumulative 2010–2017 electric vehicles | 749,000 | 366,000 | 49% |
| Total charge points | 44,300 | 13,600 | 31% |

(Population data from U.S. census; income data from U.S. Bureau of Economic Analysis; vehicle registrations from Information Handling Services (HIS) Automotive; public charging data from Alternative Fuels Data Center).

The state incorporates six of the 50 biggest U.S. metropolitan zones by population. These six territories—Los Angeles, San Francisco, Riverside, San Diego, Sacramento, and San Jose, arranged by diminishing populace—positioned among the best eight markets

in 2017 as far share of electric vehicle deals. The biggest single market by volume was Los Angeles, where territory occupants obtained in excess of 38,000 new electric vehicles in 2017, establishing more than one-fifth of the whole U.S. electric vehicle showcase [57]. Although being the state of numerous EVs, the state of California is still the home of many cities with less extensive charging infrastructure and more limited electric vehicle model availability, especially in Bakersfield, Modesto, Riverside, and Sacramento, according to [60]. In our proposal, the VSM is used as a framework for system design and analysis. In the VSM application, the EVCS is considered as the system-in-focus. In the analysis, it was apparent that the majority of the issues are linked to the alignment of System 5 (values, culture, principles, rules, and the overall policy) for all nodes. A detailed diagram of the VSM applied to the system of interest is shown in Figure 6.

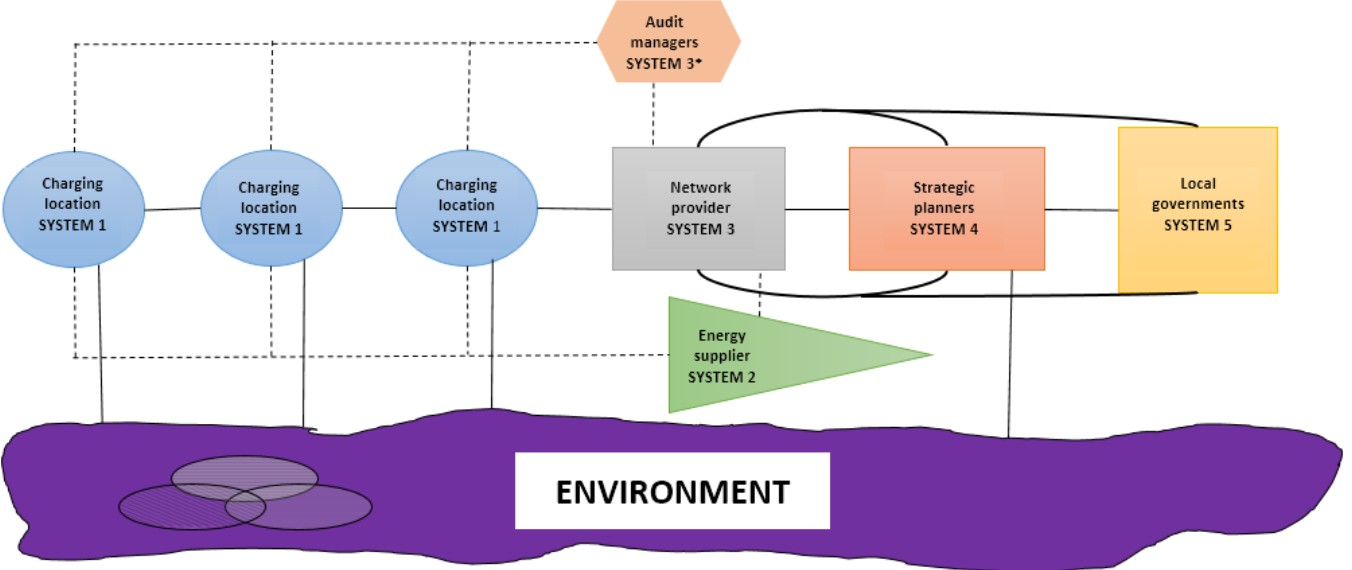

**Figure 6.** A detailed diagram of the VSM applied to the system of interest.

System 1 exists as production units, with each System 1 having its own operational management, which is in charge of directing the operational activities of the specific System 1. In this case, System 1 is the collection (system) of charging points. It functions as an autonomous unit that provides a product and has a close interaction with the environment. Each connection, simplified in Figure 1, represents a piece of information or a material exchange to monitor and control the system and to provide coordination for making decisions. These exchanges of information are represented by the different connections between the environment and the systems.

Customers rank not having enough access to effective charging stations as the third-most genuine hindrance to EV buyers, behind cost and driving range. This was indicated by McKinsey's 2016 EV study. With EV costs and ranges declining, charging could be the major obstruction. This correlation between available infrastructure and EV sales growth was verified by several studies performed by [61–63]. These correlations are comprised of multiple factors, including access to home and work charging points, the density of population, and policies and promotions. In addition, a pertinent example is the Californian EV state. In this sense, System 3 is mainly responsible for operational decisions by controlling the ongoing day-to-day performance by interpreting and ensuring the execution of System 5 policies and guidance. In this model, System 3 logically becomes the power network provider. As the System 3 power network provider offers resources and expectations for System 1, it also communicates with System 4 to reform the processes to accommodate environmental shifts.

We can recall from the VSM approach System 2 (coordination). To understand its role, we need to answer this question: Who or what is to provide for coordination within

the system? System 2 is the function that reduces or eliminates the instability caused by a conflict between the productive units (System 1s). Since charging still experiences discontinuity, conflicting information accessibility, and an absence of reliable models in many markets, educating the users is imperative. Educating users is among the most vital needs in the present market. Numerous individuals are essentially not mindful of what electric portability is, the abilities of the vehicles themselves, and why they are better (both for purchasers and for society). City pioneers have ideal stages for bringing issues to light about e-mobility.

System 3 * as the audit system, it operates to determine the source of sporadic issues by investigating and reporting to System 3 concerning the deficiencies in performance based on emergent conditions or in failure to meet accountability expectations. It is illustrated in reviewing the compliances of safety and operation standards of the charging locations.

System 4, as part of the development, investigates the nature of future development requirements. System 4 is in charge of setting up the store network to identify the conceivable changes that may emerge later on (expectation). This gives the entire framework the vital versatility to keep up its practicality after some time (e.g., shifts in an environment demanding a commensurate shift in the system). System 4 is also responsible for informing System 5 of any strategic policy implications relevant to the shifts in environmental circumstances.

Although foreseeing EV charging requests is a critical element of the system, it is important to realize the relationship between innovation and utilization of EVs. Thus, displaying and arranging are subject to introduce system vulnerabilities. In response, System 5 provides direction on an appropriate balance and prioritization between present operation and future development. System 5 is also charged with maintaining system identity by propagating the systems requirements of the EV system throughout the system. While the EV showcase is developing at a quick pace, political, financial, and innovative vulnerabilities will shape the advancement of the market in the coming years. It is imperative to target explicit, known charging needs. System 5 defines the philosophy of the whole system, in this case, the need for transparency of information and the enforcement of the roles of the stakeholders as a management principle must be highlighted as key concerns. The issue of charging foundation accessibility is intricate and vast, and building a far-reaching charging system would be cost prohibitive. Besides, because the business is developing rapidly, current momentum trends related to innovation and driver inclinations may not continue. Cities can likewise bolster one another by communicating a common foundation for a national rather than state-based approach. Campaigning for monetary help for EV programs such as those in France, the United Kingdom, and Norway, might support a more national-based strategy. Moreover, open measures for vehicle–charge point correspondence and installment may moderate a portion of these issues by empowering interoperability between charging systems, expanding advancement and rivalry, and diminishing expenses to drivers.

In this VSM analysis, we can deduce that there is ineffectiveness in System 2 coordination of productive functions. System 3 * (audit) in this case study is lacking requisite knowledge due to weakness in the identity (policy) function. Most of the literature reviews about EV optimization charging points are lacking the introduction of government and policy decision variables in the formulation of the constraints. System 5, in this sense, is problematic, providing weak clarity, communication, and alignment in the identity between the government and municipalities and other stakeholders. Table 4 presents insights from the application of the VSM, broken down by functions 1–5.

**Table 4.** A table of insights from the application of the VSM.

| | |
|---|---|
| System 1 | In this case, System 1 is the whole charging location. There might be multiple System 1s in a viable system based on the structural configuration, but they should act autonomously. |
| System 2 | Energy suppliers, standardized procedures, and cross-functional groups are some examples of stakeholders designed to accomplish System 2 functions for the EV ecosystem. |
| System 3 | In this framework, System 3 translates as the power network provider. The network provider would record the performance of System 1. The network provider also deploys policies, strategies, allocation and distribution of resources, and accountability. As an example, System 3 (network provider) administrates the working hours of employers and power source of the plug-ins, to ensure the smooth operation of the System 1 system. |
| System 3 * | For this case study, the routine audits by the compliance team or investigating the failures at the charging station to uphold the performance standards are examples of performing a System 3 * monitoring activity. |
| System 4 | Strategic planning, environmental scanning, risk identification, selection of disposal facilities, and socioeconomic trends assessment belong to the System 4 activities. All these activities would perform intelligence functions and gather information from the external environment to initiate corporate planning for the charging station. |
| System 5 | Formulating the principles and goals of the system, this function is to ensure the preservation of the system's identity as it adapts to changes that have occurred as part of the government and city's jobs. This means ensuring the autonomic management of ongoing operations and System 4, which commits to creating a balance between existing and future orientations and operational concerns of the EV ecosystem. |

## 6. Implementation in EV-Charging Locations: Discussion

One of the main challenges currently facing EV-charging locations is to improve efficiency by decreasing the charging time, and by that, simultaneously enhancing customer satisfaction. In other words, the throughput must be properly adopted within the system. Nonetheless, although the problem frame of reference suggests a more systemic consideration, it has not been viewed as a systemic issue to be understood through the perspective provided by a systems-based methodology.

The VSM plays a decisive part in the general development of a system, as it orchestrates the framework in a way that defines the systemic structural issues of managing the EV location problem from a management perspective. In the application of the VSM, the interaction of various stakeholders and the systemic structural issues can be examined. This analysis offers stakeholders and system managers an understanding to advance the system that has been developing in a fragmented reductionist fashion.

The application of this framework in EV-charging locations serves to identify systemic structural deficiencies. The primary obstacles in EV-charging location development are not singular in nature. Instead, they are interconnected and not effectively addressed in a fragmented or piecemeal fashion. It is understandable that the current state of the EV charging issue has been unresolved due to the inability to take a holistic view of the situation. Possibly a critical element is the lack of an effective accounting of System 5. As such, there is a 'rampant' lack of integrated direction in the purposes and trajectory that should be pursued.

System 4 needs to take a more effective representation of the whole system rather than just the intelligence produced by the environment. Furthermore, this framework is advising on full cooperation and reliable communication between System 5 and System 3 to reinforce the role of this later. The future mass adoption of the rechargeable electric vehicle will bring unique opportunities to consumers, automakers, and network providers. Efficient energy management solutions will have to be found for the electric vehicle to become a credible alternative to the traditional vehicle. For network providers, the widespread use of electric vehicles will create problems to be solved depending on how and when the vehicles are to be charged. The regulatory environment will have to evolve so as not to discourage suppliers from making innovative investments in the field of electric vehicles and technologies related to smarts grids and to ensure that they can integrate the consumption of these infrastructures into their curves. As an example, infrastructure upgrades may not be required if drivers can be encouraged to charge outside of

peak consumption; existing infrastructure should be able to absorb mass adoption of the electric vehicle.

A great part of the early electric vehicle charging foundation was not methodically arranged or ideally developed, or may have been permitted to evolve in a fragmented manner, with various government and private-part players using different frameworks without fundamentally holding a common vision. In numerous business sectors, this implies an electric vehicle driver needs a variety of sources, records, and information to fully understand the situation they face. This was not an issue for most early adopters of this innovation when practically all charging was done at home, and many charging stations were free. However, the "early adopter" mentality is not sufficient to move the system forward for the future.

There have been a few noteworthy endeavors directed to enhancing the client experience of the charging framework by advancing interoperability, both for drivers and for charging system administrators. California is right now formulating the Electric Vehicle Charging Open Access Act, which centers around client communication with the Electric Vehicle Supply Equipment (EVSE). This demonstration requires (1) distribution of all station areas on the Alternative Fuels Data Center (AFDC) site; (2) divulgence of all expenses before a charging occasion starts, incorporating module charges if an individual is not from the system; and (3) charge guide openness toward nonmembers of the system, including the capacity to acknowledge numerous types of installment. Actualizing these key highlights will empower more extensive access for customers. Different states, for example, Washington and Massachusetts, are additionally seeking to engage in interoperability activities. These ventures, like government support for interoperability and the utilization of open guidelines, could be vital in the long-haul development of electric vehicle charging systems.

## 7. Implication and Conclusions

As with the adoption of any new technology, the introduction of EVs into the mainstream of the transportation ecosystem is fraught with challenges and emergent difficulties. The need for alternative ways of looking at new technology integration can provide insights not accessible from more restrictive and traditional technology deployment approaches [64]. Hence, although EV technology has been developed and demonstrated as a "technical capability" to deliver an enhanced transportation technology, the deployment of this technology into the existing transportation ecosystem is much more complex than the technology itself. For instance, the existing support infrastructure and paradigm for vehicles have been built around the distribution of fossil fuel for well over a century. The result has been an efficient delivery and distribution system to support fossil-fuel vehicles (e.g., gas stations, refineries, and distribution supply chains). However, with the growing preponderance of EV technology, there is a corresponding need to think more broadly about the transportation ecosystem. Specifically, the needs that the 'new' ecosystem's requires must be developed to more effectively utilize, integrate, and capitalize on the emergent EV technology.

In this paper, we have introduced the VSM as an alternative paradigm to view EV deployment difficulties from a systems perspective. There have been several insights provided by the systemic perspective provided by the VSM for incorporating EVs into the transportation ecosystem:

1.  An holistically-structured systems inquiry: The system structure for EV deployment was identified and assessed from the holistic view of the transportation ecosystem. A rigorous process of systemic inquiry provided for insights not accessible from more "standard" non-systemic views (e.g., technology-only considerations). In effect, the VSM permitted inquiry into the whole ecosystem (technical, distribution, and utilization) rather than taking a piecemeal or fragmented approach. The result is an understanding of the entirety of EVs in relationship to the transportation ecosystem structure. This perspective ranges across a wide spectrum that includes technical, human, social, organizational, managerial, policy, and political dimensions of EVs

in relationship to the larger transportation ecosystem. The system analysis by the VSM provides the system structural understanding to more holistically develop EVs in light of the present and future transportation system.

2.  An integrated systems framework to identify systemic challenges: While there has been significant work performed on the development of EVs, there are a continuation of fragmentation and isolated development. By using a systems-based approach, provided by the application of the VSM, new and novel insights into system structural issues were generated. These insights demonstrate the ability to point to developmental challenges focused on 'whole'-system structural deficiencies. This also entails the benefit of "seeing" the fit to the existing transportation structure. By considering the entire wider array of structural issues (e.g., vehicle charging stations), existing and new structural challenges can be identified. This, in turn, can support a more integrated and coordinated developmental response to structural deficiencies impeding EV development.

3.  Identification of alternative development strategies: Given that the continuing propagation of EVs is inevitable, alternative insights can provide the basis for different development strategies. For example, infrastructure (e.g., charging stations) support might influence the investment and technology development requirements for vehicle-range requirements and vice versa. By utilizing a VSM-based system structural analysis, the opportunity to identify potential joint developmental issues was demonstrated. This can accelerate the development and more effective deployment of EV strategies. The result can be more informed decisions concerning scarce resource investment, policy development, and regulatory constraint development.

The EV revolution is here and will continue to grow into the future. Systems-based approaches, such as the VSM presented in this paper, can inform a different level of thinking for their effective integration into the existing transportation ecosystem. The more holistic analysis for the deployment of EVs offered by the alternative systems-based approaches can accelerate development. This acceleration can identify deep system structural 'blind spots' that can limit development and can suggest more efficient development strategies. New development strategies can be more appreciative of the complex nature of new technology integration into an existing transportation ecosystem. In effect, new technology can benefit from new and novel development approaches and paradigms [65]. Systems-based approaches, such as the VSM, provide an alternative to fragmented, isolated, and parochial development and deployment of new technologies.

**Author Contributions:** Conceptualization, N.U.I.H., R.J., and M.B. methodology, N.U.I.H., M.B., and C.K.; software, N.U.I.H. and M.B.; validation, N.U.I.H., M.B., and C.K.; formal analysis, N.U.I.H., M.B., and C.K.; investigation, N.U.I.H., M.B., and C.K.; resources, S.T., M.N.; data curation, N.U.I.H., M.B., and C.K.; writing—original draft preparation, N.U.I.H., M.B., and C.K.; writing—review and editing, R.J., M.N.; visualization, N.U.I.H., M.B., S.T., and C.K.; supervision, N.U.I.H., R.J., and C.K.; project administration, N.U.I.H., M.B., R.J., C.K. and M.N. All authors have read and agreed to the published version of the manuscript.

**Funding:** This research received no external funding.

**Institutional Review Board Statement:** Not applicable.

**Informed Consent Statement:** Not applicable.

**Data Availability Statement:** Data is contained within the article.

**Conflicts of Interest:** The authors declare no conflict of interest.

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
