# Peer review of "The Architecture Design of Electrical Vehicle Infrastructure Using Viable System Model Approach"

_systems, doi:10.3390/systems9010019_

Round 1

Reviewer 1 Report

This is more of a thought piece than a research paper.  It is not intended to construct an analytical model or forecast related to EVs.  However, by using the VSM framework, it does point out the need for a more holistic approach to integrating EV into the fleet and the need to think more carefully about the policies, incentives, and systems that can encourage more rapid adoption.  The paper speaks to the "S1" production system as the primary focus of much of today's activity and suggests that more attention needs to be given to the related systems (e.g., power grid, communications/coordination, policy/incentives) for EVs to become more attractive and acceptable to consumers.

I suggest that the authors make a careful editorial review of the paper to correct a few grammatical error and awkward sentences.  The writing was a bit uneven, suggesting that different authors wrote different sections of the paper and it needs someone to read the entire paper with an eye toward correct errors and improving the language in some cases.

My bottom line is that I think this is a useful paper and will contribute to advancing understanding of what is needed to promote greater penetration of EVs into vehicle market.  Perhaps it will encourage policy makers and others to think more deeply about how they can help prepare for and promote EV use.

Author Response

This is more of a thought piece than a research paper.  It is not intended to construct an analytical model or forecast related to EVs.  However, by using the VSM framework, it does point out the need for a more holistic approach to integrating EV into the fleet and the need to think more carefully about the policies, incentives, and systems that can encourage more rapid adoption.  The paper speaks to the "S1" production system as the primary focus of much of today's activity and suggests that more attention needs to be given to the related systems (e.g., power grid, communications/coordination, policy/incentives) for EVs to become more attractive and acceptable to consumers.

I suggest that the authors make a careful editorial review of the paper to correct a few grammatical error and awkward sentences.  The writing was a bit uneven, suggesting that different authors wrote different sections of the paper and it needs someone to read the entire paper with an eye toward correct errors and improving the language in some cases.

My bottom line is that I think this is a useful paper and will contribute to advancing understanding of what is needed to promote greater penetration of EVs into vehicle market.  Perhaps it will encourage policy makers and others to think more deeply about how they can help prepare for and promote EV use.

Response: Thank you for the appreciation. The entire manuscript has been proofread and edited to enhance the cohesion, clarity, and flow of the content.

Reviewer 2 Report

The authors take the brave, but very interesting approach to analyze the deployment of electrical charging infrastructure from an organizational perspective, backed by the Viable System Model (VSM) approach. I was sceptic in the beginning as I feared to be confronted with an superficial discussion. But the paper was very interesting to read due to the unusual but meaningful approach. 

Of course, the establishment of the VSM for EVCS is just the top-level structure that needs to be detailed for all 5 systems in the future. But it is a very interesting approach and valuable "step 1". I encourage the authors to further elaborate on this topic in the future and hopefully that will yield particular suggestions for all 5 involved systems (actors)!

A point for improvement: English requires spell-checking and one reference (line 204) is broken.

Author Response

The authors take the brave, but very interesting approach to analyze the deployment of electrical charging infrastructure from an organizational perspective, backed by the Viable System Model (VSM) approach. I was sceptic in the beginning as I feared to be confronted with an superficial discussion. But the paper was very interesting to read due to the unusual but meaningful approach. Of course, the establishment of the VSM for EVCS is just the top-level structure that needs to be detailed for all 5 systems in the future. But it is a very interesting approach and valuable "step 1". I encourage the authors to further elaborate on this topic in the future and hopefully that will yield particular suggestions for all 5 involved systems (actors)! A point for improvement: English requires spell-checking and one reference (line 204) is broken.

Response: Thank you so much for your valuable remarks. Definitely, we will consider to further elaborate the topic in future to grasp the broader aspects of EV. Also, the entire manuscript has been edited and necessary changes have been made where necessary. Aforementioned reference has been fixed as well.
